

# Genomic analyses provide insights into breed-of-origin effects from purebreds on three-way crossbred pigs

Yu Lin[1],*, Qianzi Tang[1],*, Yan Li[1], Mengnan He[1], Long Jin[1], Jideng Ma[1], Xun Wang[1], Keren Long[1], Zhiqing Huang[2], Xuewei Li[1], Yiren Gu[3] and Mingzhou Li[1]

[1] Institute of Animal Genetics and Breeding, College of Animal Science and Technology, Sichuan Agricultural University, Chengdu, Sichuan, China
[2] Institute of Animal Nutrition, Sichuan Agricultural University, Chengdu, Sichuan, China
[3] Sichuan Animal Science Academy, Chengdu, Sichuan, China
* These authors contributed equally to this work.

Corresponding author
Mingzhou Li,
mingzhou.li@sicau.edu.cn

## ABSTRACT

Crossbreeding is widely used aimed at improving crossbred performance for poultry and livestock. Alleles that are specific to different purebreds will yield a large number of heterozygous single-nucleotide polymorphisms (SNPs) in crossbred individuals, which are supposed to have the power to alter gene function or regulate gene expression. For pork production, a classic three-way crossbreeding system of Duroc × (Landrace × Yorkshire) (DLY) is generally used to produce terminal crossbred pigs with stable and prominent performance. Nonetheless, little is known about the breed-of-origin effects from purebreds on DLY pigs. In this study, we first estimated the distribution of heterozygous SNPs in three kinds of three-way crossbred pigs via whole genome sequencing data originated from three purebreds. The result suggested that DLY is a more effective strategy for three-way crossbreeding as it could yield more stably inherited heterozygous SNPs. We then sequenced a DLY pig family and identified 95, 79, 132 and 42 allele-specific expression (ASE) genes in adipose, heart, liver and skeletal muscle, respectively. Principal component analysis and unrestricted clustering analyses revealed the tissue-specific pattern of ASE genes, indicating the potential roles of ASE genes for development of DLY pigs. In summary, our findings provided a lot of candidate SNP markers and ASE genes for DLY three-way crossbreeding system, which may be valuable for pig breeding and production in the future.

## INTRODUCTION

Crossbreeding strategy for animals (especially agricultural poultry and livestock) is a classic and effective method aimed at improving crossbred performance (*Sevillano et al., 2016*; *Sheng et al., 2013*; *Zhang et al., 2018*). Using crossbreeding strategy, alleles that are specific to different purebreds will be inherited by offspring and thereby yield a large number of heterozygous single-nucleotide polymorphisms (SNPs) (*Sevillano et al., 2016*; *Vandenplas et al., 2016*). These heterozygous SNPs are supposed to have a contribution on

improving crossbred performance, especially in terms of growth rate, reproductive performance, production performance (egg, meat, and milk) and disease resistance (*Puppel et al., 2018*; *Ragab et al., 2016*; *Rinell & Heringstad, 2018*).

Pigs (*Sus scrofa*) were domesticated at least ~9,000 years ago and have been used as a major source of animal proteins in the human diet (*Chen et al., 2007*). There are over 730 distinct pig breeds worldwide (*Chen et al., 2007*). The worldwide distribution of pigs is dominated by six international transboundary commercial pig breeds originating in Europe, namely, Berkshire, Duroc, Hampshire, Landrace, Piétrain, and Yorkshire, of which Duroc and Hampshire pigs were developed mainly in North America (*Chen et al., 2007*). After the long-term practice for the presence of combinations of abilities, a terminal crossbreeding system with three pig breeds, namely, Duroc × (Landrace × Yorkshire) (DLY), is generally used for commercial pork production (*Hua et al., 2014*; *Liu et al., 2015*). Landrace and Yorkshire pigs share prominent traits for pork production, typically, long carcass length, thin subcutaneous fat layer, large hams, good mothering ability, and high muscularity in the carcass (*Choi et al., 2016*; *Ruusunen et al., 2012*). Duroc pigs are mainly used to enhance the growth rate and intramuscular fat in this three-way crossbreeding system (*Choi et al., 2016*). The pigs generated by this system exhibit a collection of excellent traits, such as high productivity, rapid growth, desirable pork quality and pork production (*Choi et al., 2016*).

Though previous studies have focused on allele-specific expression (ASE) genes to investigate the breed-of-origin effects from purebreds on crossbred individuals for bacterial resistance (*Wu et al., 2015*), peripheral blood (*Maroilley et al., 2017*), brain development (*Oczkowicz et al., 2018*), prenatal skeletal muscle development (*Yang et al., 2016*), adipogenesis and lipid metabolism (*Stachowiak, Szczerbal & Flisikowski, 2018*), such studies are absent for DLY three-way crossbred pigs. Here, we first estimated the distribution of heterozygous SNPs in three kinds of three-way crossbred pigs via whole genome sequencing data originated from three purebreds (including 11 Duroc, nine Landrace and 10 Yorkshire pigs). Based on this estimation, we suggested that DLY is a more effective strategy for three-way crossbreeding system among Duroc, Landrace, and Yorkshire breeds as it could generate more stably inherited heterozygous SNPs. We then sequenced a DLY pig family and identified 95, 79, 132, and 42 ASE genes in adipose, heart, liver and skeletal muscle, respectively. PCA and unrestricted clustering analysis revealed that these ASE genes were mainly tissue-specific, suggesting the potential roles of ASE genes during the development of DLY pigs. Overall, we identified a large number of candidate SNP markers and ASE genes that may have a contribution on DLY three-way crossbreeding system. These may be valuable for pig breeding and production in the future.

## MATERIALS AND METHODS

All experimental procedures and sample collection methods in this study were approved by the Institutional Animal Care and Use Committee (College of Animal Science and Technology of Sichuan Agricultural University, Sichuan, China; approval No. DKY-B20121406).

### Genome sequencing data of pigs

In this study, we first downloaded genome sequencing data of 30 pig individuals (including 11 Duroc, nine Landrace and 10 Yorkshire pigs) with mean genome coverage of ~15.47× for each individual (Table S1). We then sequenced a DLY pig family, including two grandparents (a male Landrace pig and a female Yorkshire pig), two parents (a male Duroc pig and a female LY crossbred pig) and six offspring (three male and three female DLY crossbred pigs) using Illumina HiSeq 4,000 platform (Table S2). In total, we generated ~962.46 Gb paired-end 100-bp (PE100) high quality sequencing data for 10 pig individuals, with mean genome coverage of ~38.51× for each individual (Table S2).

### SNP calling

The genome sequencing data of pigs in this study were first mapped to reference pig genome (v.11.1, see "URLs") using BWA (v.0.7.8) with default options (*Li & Durbin, 2009*). We used 'MarkDuplicates' module in package Picard (v.1.48, see "URLs") to remove duplicated reads. The module 'HaplotypeCaller' in Genome Analysis Toolkit (GATK; v.3.7) (*McKenna et al., 2010*) was used to call SNPs, which were next filtered by following criteria: QUAL < 30.0, QD < 2.0, MQ < 40.0, FS > 60.0. Sex chromosomes (X and Y) were excluded for SNP calling. To further improve the accuracy of SNP calling, we empirically depleted ~2% SNPs located in left and right tails, based on the distribution of SNP depth (Fig. S2). Finally, we identified a total of ~12.56 million (M) SNPs of three pig breeds (Table S1) as well as ~13.29 M SNPs of the DLY pig family (Table S2). We used Illumina's porcine 60K Genotyping Bead-Chip (v.2) to validate the accuracy of SNP calling for each individual of the sequenced DLY pig family (Table S3). PLINK software (*Purcell et al., 2007*) was used to calculate the mendelian error to confirm the accurate kinship of the DLY pig family.

### Calculation of the probability of heterozygous SNP

To calculate the probability of heterozygous SNP (PHS) of simulated offspring generated by different crossbreeding systems, we first calculated the frequency of three genotypes (homozygous identical with reference, homozygous distinct from reference, heterozygous) for each SNP locus based on the ~12.56M population-scale SNPs for Duroc (11 individuals), Landrace (nine individuals) and Yorkshire (10 individuals) respectively. We then simulated crossbreeding between each two pig breeds and recalculated the frequency of the same three genotypes for simulated offspring. Based on this method, we could obtain PHS for each SNP locus of the simulated offspring generated by different crossbreeding systems.

### Principal component analysis and function enrichment analyses

We performed principal component analysis with software GCTA (v.1.91.3 beta, see "URLs") based on the ~12.56M population-scale SNPs. Function enrichment analyses were performed using the online toolkit 'Metascape' with parameters 'Min Overlap = 3, *P* Value Cutoff = 0.01, Min Enrichment = 1.5' (*Zhou et al., 2019*).

## Identification of ASE genes

For each DLY individual, RNA was extracted from adipose, heart, liver and skeletal muscle. RNA-Seq libraries were prepared from total RNA using poly(A) enrichment of the mRNA to remove ribosomal RNA (rRNA). All RNA-Seq libraries were sequenced using an Illumina HiSeq4000 platform, generating a total of ~111.52 Gb transcriptome data (Fig. S3). After quality control, high-quality transcriptome data were mapped to reference pig genome (v11.1, see "URLs") using STAR software (v.2.6.0) (*Dobin et al., 2013*) with parameters "—outSAMattributes NH HI NM MD—alignEndsType EndToEnd" (Fig. S4A). The heterozygous SNPs of each DLY individual were phased via trio genotypes and only the phased exonic SNPs were used for further analyses. Reads covering multiple SNPs were trimmed to leave one SNP locus so that a read only can be counted once, a necessary step for statistical analysis. SAMtools (*Li et al., 2009*) was then used to generate a pileup file, which were used to calculate the total number of reads aligning to each allele (Fig. S4A). A total number of reads mapped to each gene for different alleles were summed up and we required a minimal count of 10 for further analysis (*Andergassen et al., 2015*; *Andergassen et al., 2017*). A binomial test was then used to assess the significance of deviation of the observed allelic biases from the expected 1:1 distribution for ASE for each sample (Fig. S4A). *P*-values were adjusted by bonferrioni method (Fig. S5). Each gene was then classified to three categories, including ASE (adjust-*P* < 0.05, binomial test), NA (not significant for ASE), no heterozygous SNP (Fig. S4A). A high-confidence ASE gene should include two ASE categories across the six DLY individuals (Fig. S4B).

## URLs

Reference pig genome, http://hgdownload.soe.ucsc.edu/goldenPath/susScr11/bigZips/susScr11.fa.gz; Picard, http://picard.sourceforge.net; GCTA, http://www.gcta-ga.org.

# RESULTS

## Distribution of heterozygous SNPs in three-way crossbred pigs

To estimate the distribution of heterozygous SNPs in three-way crossbred pigs, we calculated the PHS for simulated offspring generated by three kinds of three-way crossbreeding systems, namely, DLY, Landrace × (Yorkshire × Duroc) (LYD) and Yorkshire × (Duroc × Landrace) (YDL) (Figs. 1A–1F). This was based on the ~12.56M SNPs of the publicly available genome sequencing data of 30 pig individuals from three purebreds (including 11 Duroc, nine Landrace, and 10 Yorkshire pigs) with mean genome coverage of ~15.47× for each individual (Table S1). Compared with the other two kinds of three-way crossbreeding systems, DLY significantly exhibited more high-probability heterozygous SNPs (Fig. 1G, mean Pearson's $r = -0.99$, $P < 0.01$). This was most likely contributed to the closer genetic relationship between L and Y (Fig. 1H). As a result, compared with the other two kinds of two-way crossbreeding systems (Fig. S1A), LY significantly exhibited fewer high-probability heterozygous SNPs (Fig. S1B, mean Pearson's $r = -0.93$, $P < 0.01$). Taking together, we suggest that DLY is a more effective

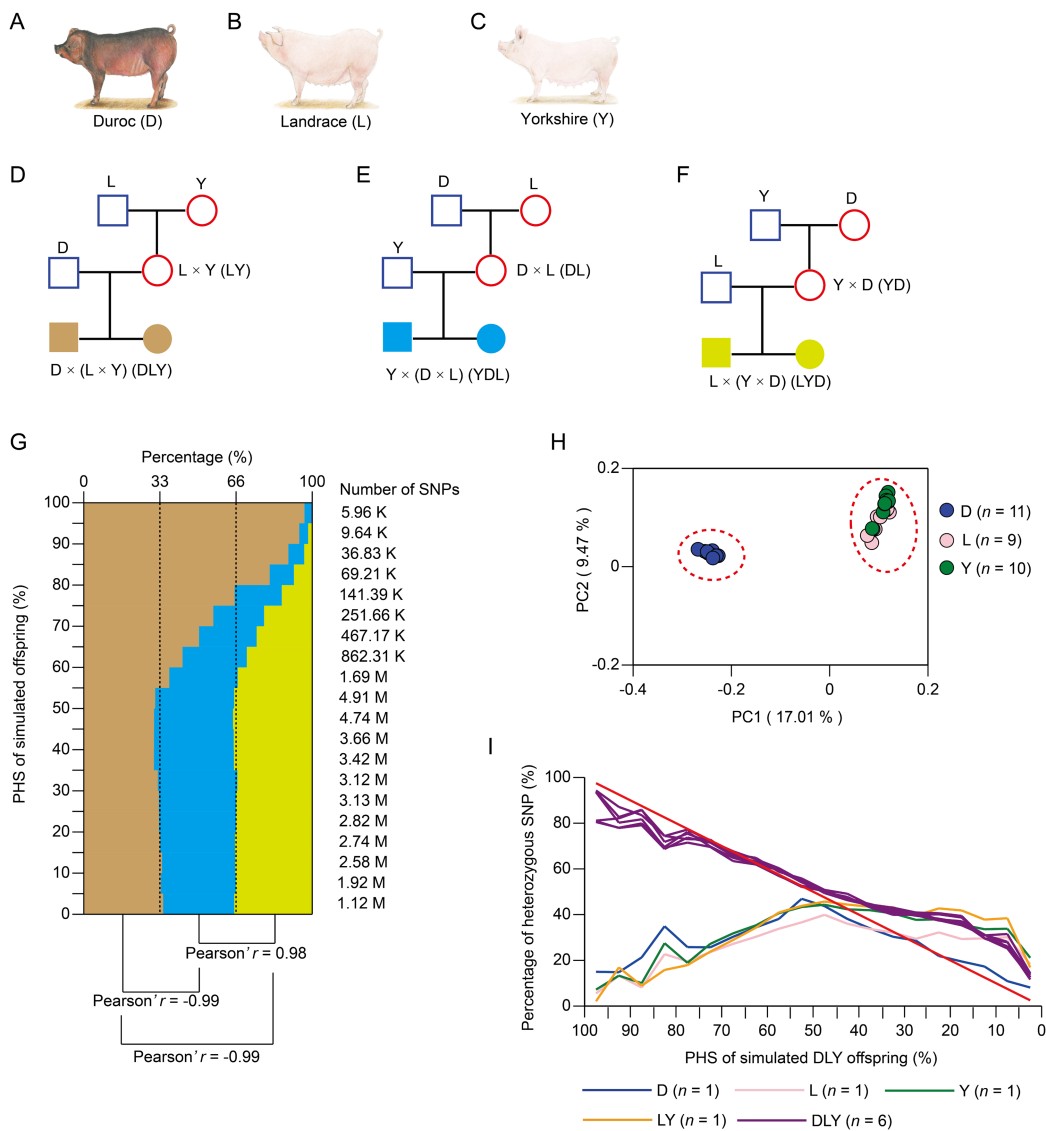

**Figure 1 Simulation of three-way crossbreeding system and validation of the accuracy of PHS.**
(A–F) Three kinds of simulated three-way crossbreeding systems among Duroc, Landrace and York-shire pigs. (G) The distribution of SNP number of simulated offspring of three kinds of three-way crossbreeding for 20 equal intervals (based on PHS, from 0% to 100%, with intervals of 5%). Pearson's *r* was inferred between each two kinds of three-way crossbreeding systems. (H) Principal component analysis (PCA) of the three pig breeds using ~12.56M population-scale SNPs. (I) Validation of PHS. Percentages of heterozygous SNPs of each pig for 20 equal intervals (based on PHS of simulated DLY offspring, from 0% to 100%, with intervals of 5%) were calculated. Pearson's *r* was inferred for each comparison between simulated DLY offspring (red line) and sequenced pig individual. (A–C) Source credit: *Li et al. (2017)*, licensed under CC BY-NC 4.0. Full-size 🖾 DOI: erj.8009/fig-1

strategy for three-way crossbreeding system among Duroc, Landrace, and Yorkshire breeds as it could yield more stably inherited heterozygous SNPs.

## Validation of the accuracy of PHS

To validate the accuracy of PHS, we sequenced a DLY pig family, including two grandparents (a male Landrace and a female Yorkshire), two parents (a male Duroc and a

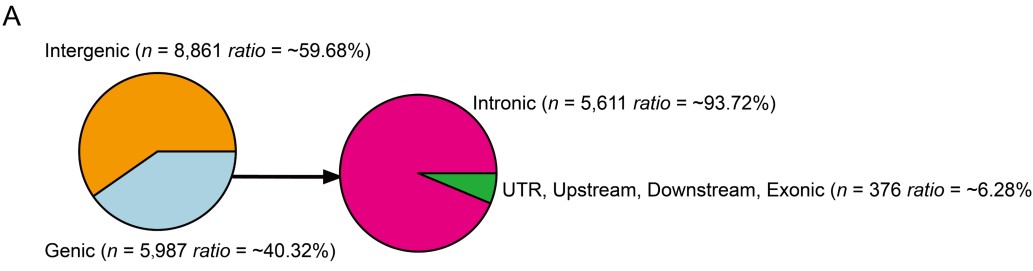

A

Intergenic (*n* = 8,861 *ratio* = ~59.68%)

Intronic (*n* = 5,611 *ratio* = ~93.72%)

UTR, Upstream, Downstream, Exonic (*n* = 376 *ratio* = ~6.28%)

Genic (*n* = 5,987 *ratio* = ~40.32%)

B

| Gene | Chromosome | Position | Genotype | Amino acid Substitution | Number of Heterozygous SNP |
|------|-----------|----------|----------|------------------------|---------------------------|
| *AHI1* | 1 | 28,589,827 | A / G | N196D | 6 |
| *AKAP9* | 9 | 72,027,468 | T / C | F3010S | 6 |
| *C8H4orf54* | 8 | 120,797,165 | T / G | I1496M | 6 |
| *ENDOU* | 5 | 78,083,854 | A / G | M21T | 6 |
| *USP20* | 1 | 270,017,455 | T / C | L104S | 6 |
| *ZNF507* | 6 | 42,281,876 | G / A | D223N | 6 |

**Figure 2 Annotation of the 14,848 high-probability heterozygous SNPs in DLY pigs.** (A) These SNPs were annotated using ANNOVAR software and classified to six genomic elements, including intergenic, intronic, UTR, downstream, upstream and exonic. (B) The detailed information of the six genes that harbored a nonsynonymous SNP were listed.

female LY crossbred individual) and six offspring (three male and three female DLY crossbred individuals) with mean genome coverage of 38.51× for each individual (Table S2). A total of ~13.29M SNPs were identified, which covered ~74.11% of the SNPs identified from the three pig purebreds as well as ~98.22% of the homozygous SNPs and ~98.36% of the heterozygous SNPs identified from Illumina's Porcine 60K Genotyping Bead-Chip (v.2) for each individual (Tables S2 and S3). As expected, the six DLY pigs exhibited a larger number of heterozygous SNPs (ranged from 6.12M to 6.26M) than the other four pigs (ranged from 4.44M to 5.62M) (Table S2). The percentage of heterozygous SNPs of each pig individual was calculated for 20 equal intervals (based on PHS of simulated DLY offspring, from 0% to 100%, at intervals of 5%) (Fig. 1I). We found that the distribution of simulated DLY offspring was significantly similar with the six DLY pigs (mean Pearson's *r* = 0.98, *P* < 0.01), but significantly different with the other four pigs (mean Pearson's *r* = −0.39, *P* < 0.01) (Fig. 1I). This result strongly confirmed the accuracy of PHS and supported our findings above.

## Function analyses of high-probability heterozygous SNPs in DLY pigs

Annotation of the 14,848 high-probability heterozygous SNPs in DLY pigs (PHS > 0.9) using ANNOVAR software (*Wang, Li & Hakonarson, 2010*) and *S. scrofa* ENSEMBL gene annotation (v.92) revealed that more than half (~59.68%) of the loci were enriched in intergenic regions (Fig. 2A), suggesting that a large fraction of the high-probability heterozygous SNPs were acted in regulatory regions of the genome (*Carneiro et al., 2014*). The majority (~93.72%) of the left loci were enriched in intronic regions, remaining ~6.28% located in upstream, downstream, UTR and exonic regions (Fig. 2A). We found

none of the coding SNPs was a nonsense, suggesting that gene loss may not play a major role for breed-of-origin effect on DLY pigs. Only six genes (*AHI1*, *AKAP9*, *C8H4orf54*, *ENDOU*, *USP20*, *ZNF507*) were identified to harbor a nonsynonymous SNP locus (Fig. 2B), indicating that very few loci had the power to completely alter protein-coding genes.

We further tested whether the six genes may have a close association with the crossbred performance. *AHI1* is required for both cerebellar and cortical development and has been previously shown to be associated with fat development and obesity via regulating insulin signaling (*Niu et al., 2012*). *AKAP9* is a member of structurally diverse proteins that have the common function of binding to the regulatory subunit of protein kinase A (*Jo et al., 2016*; *Venkatesh et al., 2016*). Mice with *AKAP9* knockout displayed decreased body fat and body weight, hematopoietic abnormalities, and an atypical plasma chemistry profile, suggesting that this gene may be important for fat and body weight development (*Gardin & White, 2011*; *Van Der Weyden et al., 2011*).

### Identification of ASE genes

To investigate the breed-of-origin effects from purebreds on gene expression pattern during the development of DLY pigs, we sequenced four representative tissues, including adipose, heart, liver and skeletal muscle (Fig. S3) and sought to identify ASE genes for each tissue. Firstly, a large fraction (ranged from ~88.59% to ~89.59%) of heterozygous SNPs for each DLY pig individual were phased via trio genotypes, that is, when at least one member of the trio was homozygous for the reference or alternate allele (*Iliadis, Anastassiou & Wang, 2012*; *Lajugie et al., 2013*) (Figs. 3A–3G). ASE genes were then identified based on these phased SNPs across the six DLY pigs for each tissue (Figs. S4A and S4B). In total, we identified 95, 79, 132 and 42 ASE genes in adipose, heart, liver and skeletal muscle, respectively, with a median value of allelic ratio ranged from 0.68 to 0.82 (Fig. 4A; File S2). Notably, we observed that ~78.09% (196/251) of ASE genes were only detected in one tissue (Fig. 4B), indicating that ASE genes were mainly tissue-specific, consistent with the results of principal component analysis (Fig. 4C) and unrestricted cluster analysis (Fig. 4D) based on gene expression of the 251 nonredundant ASE genes.

### DISCUSSION

In this study, we investigated the breed-of-origin effects from purebreds on DLY three-way crossbred pigs in genomic and transcriptional levels. In total, we identified 251 ASE genes across adipose, heart, liver and skeletal muscle. To further explore the characteristics of these ASE genes, we made a comparison with ASE genes reported in previous studies for pigs (*Esteve-Codina et al., 2011*; *Maroilley et al., 2017*; *Oczkowicz et al., 2018*). We found none of the ASE genes was shared across the four datasets, suggesting that ASE gene may be tissue-specific or breed-specific (Fig. S8). Among the 251 nonredundant ASE genes, only 38 (~15.14%) were covered by other datasets, remaining 213 (~84.86%) were first reported in this study (Fig. S8).

The metabolism of adipose is deeply associated with pork production and quality, which are the two major commercial traits for pig industries. In adipose, the top thee significant

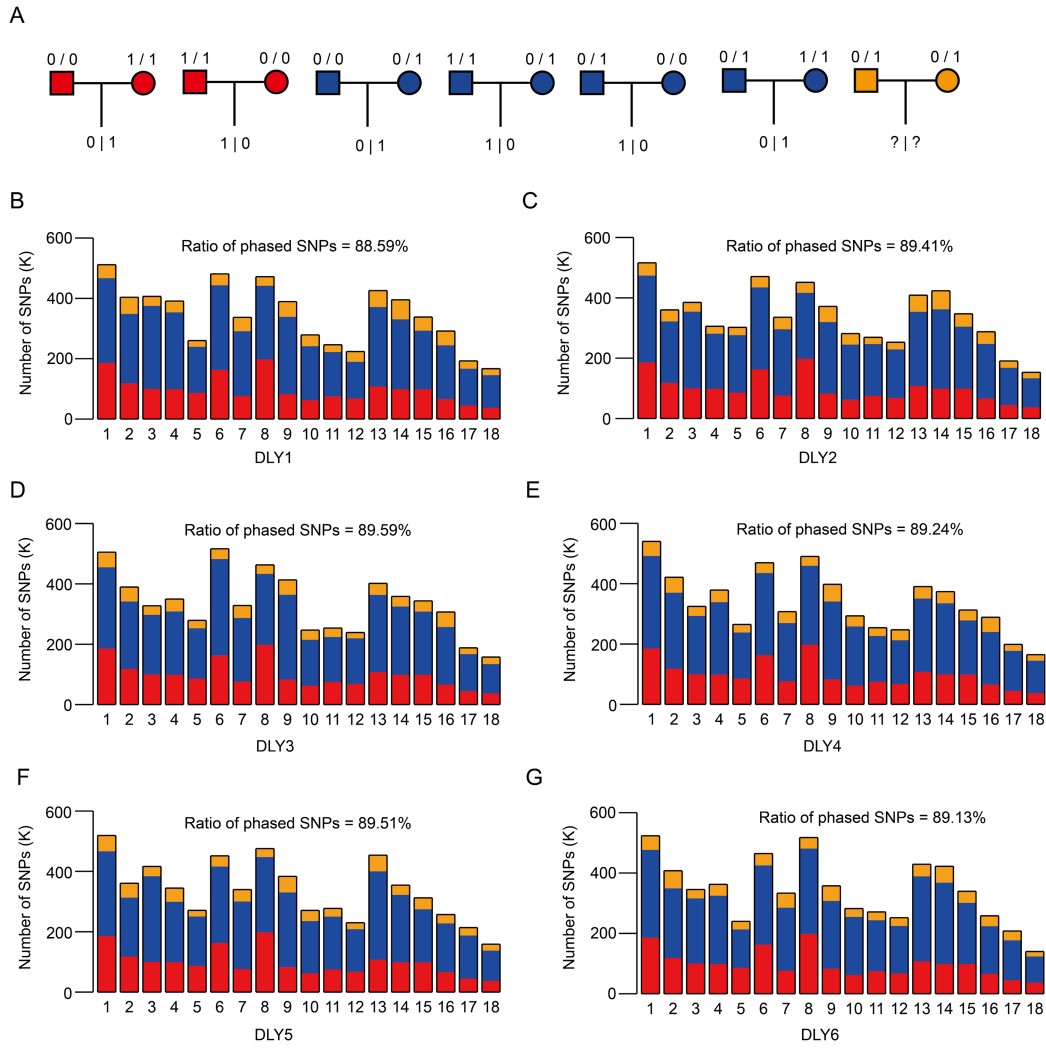

**Figure 3  SNP phasing of the six DLY pigs via trio genotypes.** (A) The theory for phasing SNP using trio genotypes. For each heterozygous SNP locus of offspring, it could be inherited by parents based on seven conditions, among which two (refers to red, where parents are both homozygous but distinct) and four (refers to blue, homozygous for one parent and heterozygous for the other parent) can be used to phase. (B–G) The distribution of phased SNPs across autosomes of the six DLY pigs, ranged from 88.59% to 89.59%.                               

GO pathways of ASE genes include 'fatty acid metabolic process' ($P = 7.24 \times 10^{-9}$), 'acylglycerol metabolic process' ($P = 6.31 \times 10^{-8}$) and 'regulation of cholesterol metabolic process' ($P = 8.13 \times 10^{-7}$) (Fig. S9A; Table S4), suggesting that ASE genes may play important roles in adipose development for DLY pigs. For example, *FABP* is supposed to participate in metabolism of long-chain fatty acids and has been proved to have a close relationship with carcass back fat thickness and intramuscular fat contents (*Cho et al., 2011*). *VIM* encodes a type III intermediate filament protein and is required for the normal accumulation of body fat (*Wilhelmsson et al., 2019*). *ACSL1* encodes an isozyme of the long-chain fatty-acid-coenzyme A ligase family and functions in converting free long-chain fatty acids into fatty acyl-CoA esters; a previous study suggested this gene

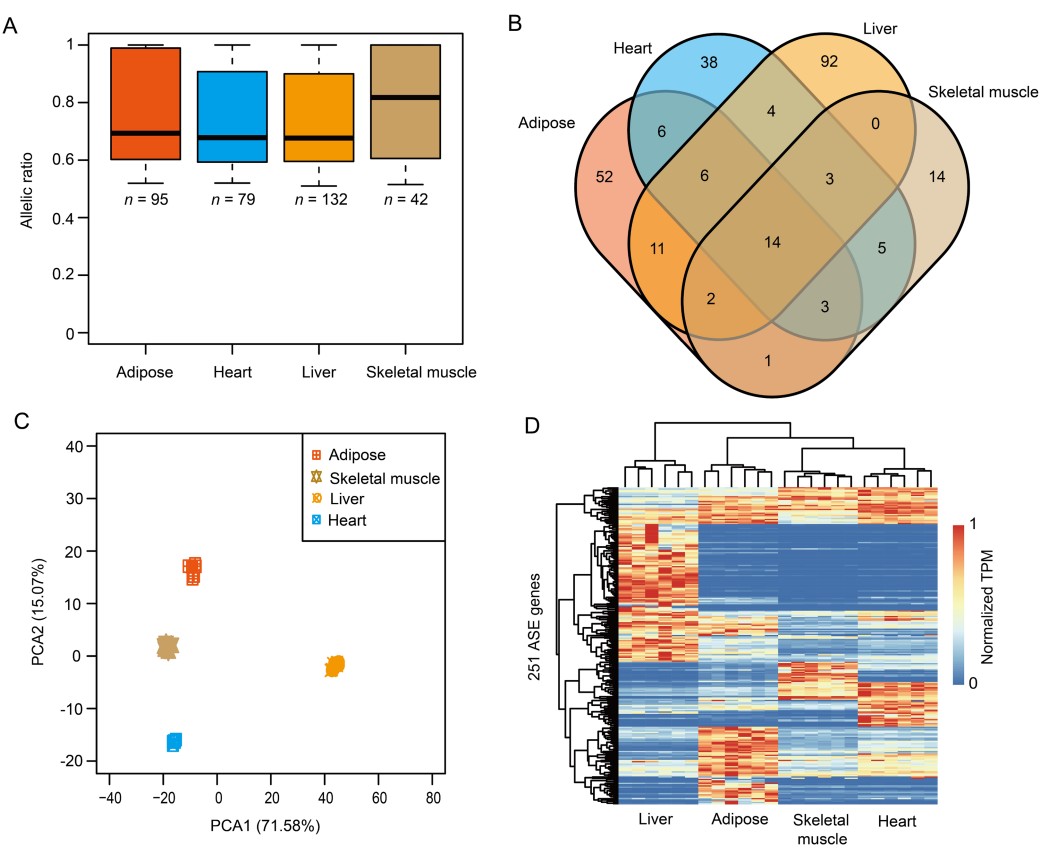

**Figure 4 The results of ASE genes.** (A) A total of 95, 79, 132 and 42 ASE genes were identified in adipose, heart, liver and skeletal muscle, with a median value of allelic ratio ranged from 0.68 to 0.82. (B) The venn diagram of ASE genes among adipose, heart, liver and skeletal muscle. (C) Principal component analysis (PCA) of the 251 nonredundant ASE genes. (D) Heatmap of the 251 nonredundant ASE genes.

might contribute to the capacity of fat deposition and meat quality in pig breeds (*Li et al., 2012*).

Liver is the major organ responding to synthesis and degradation of many important metabolites, like proteins, fatty acids, carbohydrates, drugs and bile acids. In liver, ASE genes were enriched in 18 GO and KEGG pathways, such as 'Glutathione metabolism' ($P = 2.88 \times 10^{-8}$), 'steroid metabolic process' ($P = 1.82 \times 10^{-7}$), 'monocarboxylic acid metabolic process' ($P = 2.95 \times 10^{-7}$), 'small molecule catabolic process' ($P = 2.82 \times 10^{-6}$), 'Valine, leucine and isoleucine degradation' ($P = 2.14 \times 10^{-5}$) (Fig. S9C; Table S6).

The GO and KEGG pathways for ASE genes in heart and skeletal muscle were associated with muscle development, including 'regulation of muscle contraction' ($P = 3.09 \times 10^{-9}$), 'muscle system process' ($P = 8.13 \times 10^{-6}$), 'muscle structure development' ($P = 6.76 \times 10^{-4}$), 'response to calcium ion' ($P = 4.07 \times 10^{-3}$) and 'Tight junction' ($P = 5.89 \times 10^{-3}$) (Figs. S9B and S9D; Tables S5 and S7).

Taken together, we speculate that ASE genes may play important roles for the development of DLY pigs, indicating the potential influence of breed-of-origin effects from purebreds on gene expression of crossbred individuals.

## CONCLUSION

In this study, we simulated three kinds of three-way crossbreeding system among Duroc, Landrace and Yorkshire breeds, and estimated the distribution of heterozygous SNPs in terminal crossbred pigs via whole genome sequencing data from three purebreds. We found DLY three-way crossbreeding system could yield more stably inherited heterozygous SNPs in the terminal crossbred offspring, suggesting that DLY is a more effective strategy for three-way crossbreeding. We also sequenced a DLY pig family and identified a lot of ASE genes in adipose, heart, liver and skeletal muscle. These ASE genes exhibited a tissue-specific pattern and were involved in many important functional pathways, indicating the potential influence of breed-of-origin effects from purebreds on gene expression of DLY crossbred pigs. These discoveries may provide a valuable resource for pig breeding and production in the future and set an excellent model of crossbreeding analysis for other agricultural animals.

## ABBREVIATIONS

| | |
|---|---|
| **ASE** | allele-specific expression |
| **SNP** | single-nucleotide polymorphisms |
| **PHS** | probability of heterozygous SNP |
| **DLY** | Duroc × (Landrace × Yorkshire) |
| **LYD** | Landrace × (Yorkshire × Duroc) |
| **YDL** | Yorkshire × (Duroc × Landrace) |
| **PCA** | principal component analysis |
| **QUAL** | variant quality |
| **QD** | variant confidence/quality by depth |
| **MQ** | mapping quality |
| **FS** | Phred-scaled $p$-value using Fisher's exact test to detect strand bias. |

### Funding

This work was supported by grants from the National Key R & D Program of China (2018YFD0500403), the National Natural Science Foundation of China (31802044, 31872335 and 31772576), the Sichuan Province & Chinese Academy of Science of Science & Technology Cooperation Project (2017JZ0025), the Science & Technology Support Program of Sichuan (2016NYZ0042 and 2017NZDZX0002), the Earmarked Fund for China Agriculture Research System (CARS-35-01A) and the China Postdoctoral Science Foundation (2018M643514). The funders had no role in study design, data collection and analysis, decision to publish, or preparation of the manuscript.

### Grant Disclosures

The following grant information was disclosed by the authors:
National Key R & D Program of China: 2018YFD0500403.
National Natural Science Foundation of China: 31802044, 31872335 and 31772576.

Sichuan Province & Chinese Academy of Science of Science & Technology Cooperation Project: 2017JZ0025.
Science & Technology Support Program of Sichuan: 2016NYZ0042 and 2017NZDZX0002.
China Agriculture Research System: CARS-35-01A.
China Postdoctoral Science Foundation: 2018M643514.

## Competing Interests

The authors declare that they have no competing interests.

## Author Contributions

- Yu Lin conceived and designed the experiments, performed the experiments, analyzed the data, contributed reagents/materials/analysis tools, prepared figures and/or tables, authored or reviewed drafts of the paper, approved the final draft.
- Qianzi Tang performed the experiments, analyzed the data, contributed reagents/ materials/analysis tools, prepared figures and/or tables, approved the final draft.
- Yan Li analyzed the data, contributed reagents/materials/analysis tools, prepared figures and/or tables, authored or reviewed drafts of the paper, approved the final draft.
- Mengnan He performed the experiments, prepared figures and/or tables, authored or reviewed drafts of the paper, approved the final draft.
- Long Jin performed the experiments, prepared figures and/or tables, authored or reviewed drafts of the paper, approved the final draft.
- Jideng Ma performed the experiments, prepared figures and/or tables, authored or reviewed drafts of the paper, approved the final draft.
- Xun Wang performed the experiments, authored or reviewed drafts of the paper, approved the final draft.
- Keren Long performed the experiments, authored or reviewed drafts of the paper, approved the final draft.
- Zhiqing Huang performed the experiments, authored or reviewed drafts of the paper, approved the final draft.
- Xuewei Li performed the experiments, authored or reviewed drafts of the paper, approved the final draft.
- Yiren Gu performed the experiments, authored or reviewed drafts of the paper, approved the final draft.
- Mingzhou Li conceived and designed the experiments, contributed reagents/materials/ analysis tools, prepared figures and/or tables, authored or reviewed drafts of the paper, approved the final draft.

## Animal Ethics

The following information was supplied relating to ethical approvals (i.e., approving body and any reference numbers):

All studies involving animals were conducted according to Regulations for the Administration of Affairs Concerning Experimental Animals (Ministry of Science and Technology, China, revised in June 2004). All experimental procedures and sample collection methods in this study were approved by the Institutional Animal Care and Use

Committee of the College of Animal Science and Technology of Sichuan Agricultural University, Sichuan, China, under permit No. DKY-B20121406. Animals were allowed free access to food and water under normal conditions, and were humanely sacrificed as necessary, to ameliorate suffering.

## Microarray Data Deposition

The following information was supplied regarding the deposition of microarray data:

Microarray data is available at NCBI GEO: GSE123327.

## Data Availability

The whole-genome sequencing data of ten pig individuals and transcriptome data of the six DLY pig individuals of the DLY family are available at NCBI BioProject: PRJNA507853.

The genotyping data of the Illumina's porcine 60K Genotyping Bead-Chip (v.2) are available at the NCBI Gene Expression Omnibus (GEO): GSE123327.

## Supplemental Information

Supplemental information for this article can be found online at http://dx.doi.org/10.7717/peerj.8009#supplemental-information.

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
