# Peer review of "Genomic analyses provide insights into breed-of-origin effects from purebreds on three-way crossbred pigs"

_PeerJ, doi:10.7717/peerj.8009_

## Round 0.1 · original submission · Major Revisions

As you can see from the reviewer reports, one reviewer (R2) was more critical than the other and both reviewers suggested a few points to be clarified and considered, and hence I have decided for 'Major Revision'.

Reviewer 1 ·

Basic reporting

I think the paper is interesting and worthy publishing in PeerJ. Nevertheless the style of language could be improved. Introduction section does not contain referneces of recent papers on ASE in pigs. Some sentences need to be clarified(for example abstract - line 21 -
three kinds of crossing should be listed,lines 47-48 - I think it depends on country)

Experimental design

Methods section - not enough information is given. For example - lack of the database from which the genomes were downloaded, lack of the details of RNA-seq library preparation. Some short cuts should be explained _ QD,MQ, FS. How was the functional enrichment analysis performed, with which dataset?

Validity of the findings

Dicussion section is lacking references to previous work on ASE in pigs and other mammals (estimation of percentage of ASE would be helpful)

Annotated reviews are not available for download in order to protect the identity of reviewers who chose to remain anonymous.

Reviewer 2 ·

Basic reporting

The manuscript is written in a clear and unambiguous English. The introduction obviously describes the background of the study, however the research hypothesis should be clearly stated. The quality of presented data is high.

Experimental design

The experiments have been conducted in conformity with ethical standards and within the aims and scope of the journal. The results of such a study may potentially be helpful to unravel the genetic background of traits important in pig industry, especially by indicating genes modulated by cis-regulatory factors. The methods described in the study allow to replicate the experiment. However, some information is missing:
- Was the binominal test also performed for data obtained after whole genome sequencing to check for the biases in variant calling at the level of genomic DNA? Such a normalization step is a standard procedure during validation by more accurate methods and allows to reduce false positive results.
- Did authors consider only genes in which all SNPs were found to be ASE SNPs or only a single ASE SNP was sufficient to indicate a gene as imbalanced?
- Were the expression data adjusted for multiple testing?

Validity of the findings

My major concern regarding this manuscript is that the major scope of this study was to detect ASE by global RNA sequencing but the validation step by a more accurate method (e.g. by pyrosequencing) was not conducted. At least several genes (other than imprinted ones) should be validated to confirm subtle changes in expression of both alleles as observed after analysis of RNA seq data.
The authors concluded several times that the ASE genes detected play a major role during development of DLY pigs. In my opinion, such a conclusion is exaggerated especially that authors only detected such genes without performing any functional study. Their investigation revealed that such a phenomenon exists and is more common for genes expressed in adipose tissue (similar results were published by Schachtschneider et al., 2015 BMC Genomics. 2015;16:743) and significant cis-regulatory factors are involved but it is not possible to tell which genes are truly involved in development of the studied pigs. This is more a speculation rather than a conclusion.
Taking above into consideration the manuscript should not be published in its present form without any validation of allelic ratios for at least several genes by a more accurate method.

Additional comments

No comment

---

## Round 0.2 · accepted · Accept

The reviewers do not have any further comments and also agreed with the answers you gave in your previous reply. I do not have either any comments for further improvements, hence I decided to 'Accept' your manuscript for publication.

Reviewer 1 ·

Basic reporting

no comments

Experimental design

no comments

Validity of the findings

no comments

Additional comments

no comments

Reviewer 2 ·

Basic reporting

No comment

Experimental design

No comment

Validity of the findings

No comment

Additional comments

The authors have significantly improved their study and implemented changes in text. Although they did not meet all my suggestions, their explanation is adequate. The manuscript is now suitable for publication in PeerJ.